# A Telehealth Intervention for Nutritional Counseling in Amyotrophic Lateral Sclerosis Patients

**DOI:** 10.3390/jcm11154286

**Published:** 2022-07-23

**Authors:** Fabiola De Marchi, Marcella Serioli, Alessandro Collo, Evelyn Gisell Belotti, Francesca Alloatti, Giampaolo Biroli, Andrea Bolioli, Roberto Cantello, Sergio Riso, Letizia Mazzini

**Affiliations:** 1ALS Centre, Department of Neurology, Maggiore della Carità Hospital, University of Piemonte Orientale, 28100 Novara, Italy; 20022398@studenti.uniupo.it (E.G.B.); roberto.cantello@med.uniupo.it (R.C.); letizia.mazzini@uniupo.it (L.M.); 2Clinical Nutrition and Dietetics Unit, Maggiore della Carità Hospital, 28100 Novara, Italy; marcella.serioli@gmail.com (M.S.); alessandro.collo@maggioreosp.novara.it (A.C.); giambiro@libero.it (G.B.); sergio.riso@maggioreosp.novara.it (S.R.); 3H-FARM Innovation, 10121 Torino, Italy; francesca.alloatti@h-farm.com (F.A.); andrea.bolioli@h-farm.com (A.B.)

**Keywords:** neurodegenerative diseases, telemedicine, patients’ management

## Abstract

Nutritional status is one of the most relevant prognostic factors in Amyotrophic Lateral Sclerosis (ALS), and close monitoring can help avoid severe weight loss over the disease course. We describe the impact of a Chatbot webapp on improving the communications between physicians, patients, and/or caregivers for dietary monitoring. We developed a chatbot that provides patients with a tool to register their meals through an intuitive and carefully designed conversational interface. Patients recorded their dietary intake twice weekly and received an adequate nutritional recommendation monthly. We monitored their functional and nutritional parameters. The data were compared with a control group followed up by standard counseling. We enrolled 26 patients. Regarding feasibility, 96% of participants completed the three-month follow-up, and 77% ended the six months. Regarding the change in weight in the Chatbot group, we observed a weight stabilization (F = 1.874, *p*-value: 0.310 for changes) over the telehealth compared to the control group (F = 1.710, *p*-value: 0.024 for changes). A telehealth approach for nutritional support is feasible and reproducible in an ALS setting: frequent monitoring turned out to help prevent further weight loss, allowing an early nutritional strategy adjustment.

## 1. Introduction

Amyotrophic Lateral Sclerosis (ALS) is a rare neurodegenerative disease that leads progressively to physical immobility; therefore, the patient faces an inevitable loss of independence [1]. Thus, new technologies such as mobile healthcare applications could represent a valid support for patients and caregivers, especially when the first symptoms begin in relatively young patients that are professionally active and receptive to digital innovations [2]. A multidisciplinary specialist team approach is the gold standard of ALS management for improving survival and quality of life across symptom management [3]. Nutritional status is one of the most relevant prognostic factors for ALS since malnutrition is common, and negatively affects prognosis [4,5] by exacerbating muscle weakness and compromising the respiratory and immune systems [6,7]. Weight loss is persistent during the disease [8], and is related to several factors, such as dysphagia, inappetence, depression, mobility difficulties, and changes in energy homeostasis [9]. Therefore, nutritional status and dysphagia assessment is recommended at diagnosis and during the follow-up every 3 months, to detect early malnutrition and/or dysphagia and to perform an appropriate feeding strategy in terms of consistency, caloric-nutritional intake, and safety and effectiveness of nutrition.

Our Centre standard of care requires a nutritional in-visit assessment quarterly. We hypothesized that tighter nutritional control, using a dedicated mobile application, could positively impact maintenance of the patient’s body weight [10]. In a previous study we demonstrated the feasibility of telemedicine in ALS patients during the COVID-19 pandemic, observing a reduction in the slowdown of functional measures and an improvement in anxiety and depression scales [11,12]. This study evaluated the eventual positive effects of nutritional counseling through a dedicated e-health Chatbot (a webapp application) on improving the communication between healthcare providers and ALS patients for dietary and functional outcome monitoring compared to a group of matched patients followed-up with standard multidisciplinary care.

## 2. Materials and Methods

### 2.1. Standard Protocol Approval, Registrations, and Patient Consents

The Ethics Committee approved the study at Maggiore della Carità Hospital, Novara, Italy (number CE19/20; 13 March 2020). For this project, written informed consent was obtained from all participants.

### 2.2. Study Design and Aim

A pilot clinical trial with telehealth support was conducted to maintain body weight and avoid malnutrition in ALS participants. The primary outcome aimed to test the feasibility, safety, and tolerability of telehealth with a dedicated e-health Chatbot, comparing the results with those obtained from a control group of ALS patients followed-up with a standard of care.

The research questions were the following:(1)Is dietary monitoring through a Chatbot feasible with ALS patients?(2)Is the Chatbot approach with a more intensive dietary control effective in stabilizing the body weight of ALS patients?(3)Does the improvement in dietary parameters also influence a functional and quality of life outcome monitored by the ALS functional rating scale-revised score scale (ALSFRS-R [13,14,15]) and the ALS Assessment Questionnaire ALSAQ-40 [16]?

All included patients were evaluated for a maximum period of six months by team members both in telemedicine and by out-patient visits with:(1)neurological and functional evaluation (ALSFRS-R), monitoring of intercurrent events;(2)dietary evaluation by diet monitoring (e.g., collection of weight, body mass index (BMI), caloric intake, and diet consistency);(3)evaluation of the quality of life by collecting ALSAQ-40?

### 2.3. Participants

Recruitment and enrollment took place at the Tertiary ALS Center at Maggiore della Carità Hospital, Novara, Italy, between 1 May 2020 and 31 June 2021. We included patients:(1)with ALS based on El Escorial criteria [17,18];(2)resident in Italy;(3)age > 18 years;(4)who were able to understand the study protocol and able to provide informed consent;(5)with the presence of caregiver if necessary;(6)who were able to manage personal computers or smartphones, and internet connection for data sharing.

Patients with severe cognitive decline who were unable to consent were excluded from the study. For each patient, we collected demographic and clinical information: diagnosis, type of onset, disease duration, ALSFRS-R and BMI at baseline (T0) and monthly for six months, and intercurrent events. As control group, we identified 20 consecutive patients followed-up with standard nutritional counseling between 2017 and 2018, and matched with the Chatbot group for clinical and demographic features.

### 2.4. Interventions

The proposed technological device consisted of a chatbot (developed by H-FARM Innovation) that guided the patients throughout the digital version of the dietary intake diary for diet management. The telemedicine approach was performed using the online platform “IoMT Connected Care Platform (Ticuro Reply)”, where we generated a personal account for each healthcare provider and patient/caregiver. We developed a textual chatbot that allowed patients to fill out their dietary diaries intuitively and asynchronously. However, due to the well-known speech difficulties in many ALS patients, the interface did not include speech components as voice is a problematic element for ALS patients. To allow the chatbot use in subjects with reduced mobility, we created an interface that was usable with the exclusive use of buttons that do not require typing. After physician, dietician, and speech therapist assessment, a personalized, flexible, normo-caloric dietary plan was performed and discussed, with respect to total energy expenditure, estimated through the rapid formula 30 kcal/kg body weight, with or without oral nutritional supplements administration, as appropriate. Different dietary plans based on food consistency were available according to swallowing ability (regular, soft, minced, pureed, and semi-liquid texture). Therefore, a patient may shift the texture of diet and intensity/route of nutritional intervention over time. Dietary intake was registered twice weekly through an interactive diary (both for food and beverage) and subsequently calculated by a trained dietician. The data extracted from each interaction was reported in a readable format (pseudo-anonymized Excel file). The data were used to process and monitor the patients’ nutritional clinical progress and, consequently, adjust monthly the type of diet to be proposed. Data from the Chatbot group were compared with data collected in ALS patients followed by standard multidisciplinary care between 2017 and 2018, matching them by gender, age, and disease duration of enrollment (or “monitoring onset”).

### 2.5. Outcome Measures

Safety, tolerability, and device compliance were considered as primary outcomes. All reported adverse events (during out-patient visits, by telephone, or email) were considered and registered. Tolerability and compliance with the e-health intervention were calculated as the percentage of participants who completed at least 80% of counseling sessions. Tolerability and compliance to dietary plans were defined as consuming at least 75% of the recommended intake (calculated as a monthly mean value based on two day diaries) and adherence to prescribed food textures. As for efficiency, we considered weight maintenance/gain as an outcome. Patients measured body weight monthly at home by a personal scale, and at every in-person visit (at baseline, after three and six months) by a mechanical column scale (SECA 700) with a sensitivity of 0.1 kg, as a part of the multidisciplinary standard of care. Height was measured at the baseline by the same device with a sensitivity of 0.5 cm A secondary efficacy outcome was obtained by monitoring both groups’ ALSFRS-R score slowdown. The tertiary efficacy outcome, evaluated only in the Chatbot group, was collected by monitoring the changes in the ALSAQ-40 score over the telehealth trial. These outcomes were assessed after completion of the three and six-month intervention.

### 2.6. Statistical Analysis

Variables were summarized as mean (and standard deviations), frequencies, and percentages as appropriate. Categorical variables were compared by Chi-squared test; the t-student ANOVA test compared continuous variables. For continuous longitudinal measures (before and after Chatbot use), we used linear mixed factor analysis of variance to evaluate the effect of the intervention over time on primary and secondary clinical outcomes. In addition, we added a Tukey multiple comparisons test as a post-hoc test. The same statistical approach was used for weight and ALSFRS-R score. We did not calculate a target sample size as a preliminary pilot investigation study, Data analysis was conducted with IBM SPSS Statistics for Windows, Version 25.0 (IBM Corp., Armonk, NY, USA).

## 3. Results

### 3.1. Study Population

We screened 30 consecutive ALS patients based on the inclusion criteria and enrolled 26 participants. Four patients were excluded from the trial: two for rapid deterioration with death in the following month, one for tracheostomy, and one for poor internet network. The results were compared to data obtained by 20 consecutive patients followed up with standard nutritional counseling between 2017 and 2018. The demographic and clinical variables of the two groups at diagnosis and baseline are shown in Table 1. For the first group, at diagnosis, the mean ALSFRS-R score 1–3 was 11.35 (SD: 0.94), score 4–6 was 9.73 (SD: 1.82), score 7–9 was 8.58 (2.44), and score 10–12 was 11.54 (11.27). Fourteen patients (54%) had a normal caloric intake, 12 (46%) had a hypocaloric intake, and none had a hypercaloric intake. The daily caloric intake suggested was 1920 Kcal (SD; 280). Twelve patients (46%) had a regular texture diet, and 14 had a creamy diet (54%).

### 3.2. Feasibility and Compliance with Counseling

In the Chatbot group, 25/26 (96%) participants completed the three-month follow-up, and one was lost at the follow-up. Finally, 20/26 (77%) ended the six-month follow-up (two clinically worsened, and four were not accurate in filling in the chatbot diary and therefore were not considered as completed).

### 3.3. Efficacy

Comparing change in weight in the Chatbot group before and during the technology use, we observed a weight stabilization-with a slight initial increase-over the telehealth monitoring (67.80 kg at diagnosis, vs. 66.27 kg at baseline and 67.66 kg at month 6; F = 1.874, *p*-value: 0.310; Figure 1, left panel). In contrast, we observed a significant progressive slowdown in weight in the control group (65.55 kg at diagnosis, vs. 63.98 kg at baseline and 62.58 kg at month 6; F = 1.710, *p*-value: 0.024; Figure 1, right panel).

Four patients had gastrostomy for severe dysphagia during the Chatbot monitoring. 13 patients added dietary supplements (50%). In a comparative mixed model, with an inter-rate analysis, we obtained a difference that trends towards significance (*p*-value = 0.070; Figure 2) between the Chatbot group and the control group.

### 3.4. Secondary Outcomes

ALSFRS-R: as shown in Figure 3, regarding other secondary outcomes, we did not observe a significant reduction in the slowdown of ALSFRS-R between baseline, month 3 and month 6 (27.65 at month 3, and 24.50 at month 6; *p*-value: 0.001). Similarly, we observed similar progression in the control group (30.95 at month 3 and 26.26 at month 6; *p*-value: 0.001). The analysis of the ALSFRS-R sub-scores (item 1–3, 4–6, 7–9 and 10–12) also showed a linear decline in all items.

ALSAQ-40: Anecdotal feedback on the quality of life during the dietary intervention was collected using the ALSAQ-40 scale. At the end of the intensive nutritional monitoring, the global scores for physical mobility, activities of daily living and independence, and eating and drinking increased significantly, signifying a clinical worsening despite the treatment (respectively *p*-values: 0.004, 0.001, and 0.001). On the contrary, during the six months of follow-up, we observed stabilization of “communication area” (*p*-value: 0.447) and a relative increase reduction for the “emotional reactions area” (*p*-value: 0.070 for baseline to week 8, 0.914 from week 8 to 16 and 0.606 from week 16 to week 24) (Figure 4).

## 4. Discussion

These results demonstrated that intensive monitoring of nutritional parameters in ALS patients is feasible and incisive in dietary and weight control, and Chatbot is likely to be useful in providing frequent dietary monitoring and advice ALS patients. Primarily, considering the disease-related disabilities and difficulties, we observed high compliance in the chatbot use, which also meant good feedback from patients and caregivers for food education. In addition, participation in periodical nutritional counseling supported by a Chatbot webapp showed stabilization of body weight compared with a control group followed up with standard multidisciplinary care (quarterly out-patient visits). However, we did not observe a reduction in the ALSFRS-R score slowdown in the Chatbot app group compared to the control group. The use of a tool potentially able to slow down the weight decline in ALS patients in a relatively long time (6 months) is undoubtedly encouraging and worthy of further investigation.

As widely discussed in the literature, the role of BMI in disease progression and survival is prevalent, suggesting that high-energy reserves in the first disease phases can mitigate the increased energy demands occurring over the disease course. Several studies investigated the role of BMI in disease progression and survival: the common evidence suggests that a higher premorbid BMI is related to a better functional outcome [19,20]; in addition, patients with moderate obesity (BMI 30–35) have more prolonged survival [21,22].

A similar approach to our study was previously used by Willis et al. in 2019 [23], where three different dietary monitoring approaches were evaluated: “standard care” by a physician or nurse, “in-person” counseling with a specialized dietician, or counseling supported by a “mHealth app”: an increment of the caloric intake was observed with the e-health approach, compared to the standard care, even without a significant difference in body weight in the three groups.

Conventionally, in Our Centre, as multidisciplinary care, all patients with motor neuron diseases, independently of their needs and metabolic status, undergo counseling with a dedicated dietologist and dietician, so it was not possible to compare the three approaches. Based on our standard of care, we developed a structured telehealth intervention managed by a dietician (that never changed during all the trials to avoid treatment bias). Positive results of this method support the role of specialized healthcare providers in this regard and the relevance of close-range monitoring. In addition, previously, we obtained positive results from telemedicine and telehealth approaches during the COVID-19 pandemic [12], allowing us to replicate and implement this procedure within our multidisciplinary standard of care.

Several strengths of this proposed study should be highlighted. We included 26 consecutive patients with mean disease duration (from diagnosis) of about one year, but necessarily with different phenotype and disease stages due to the implicit variability of the disease in each patient (indeed, in our analyzed cohort ALSFRS-R scores ranged from 13 to 44/48 at baseline). The patients’ variability and the low rate of drop-out in chatbot use support the idea that this technology is feasible and reproducible in most patients. The only patients’ group in which we believe this support may be less helpful is that of patients with percutaneous gastrostomy, where changes in nutritional support are doubtlessly less frequent. The ease and reproducibility of the technology can allow lower access of patients in the hospital setting, increasing duration within the visits. This approach could also be easily reproduced for clinical trial monitoring, where close monitoring of the clinical and functional patient parameters is required.

The primary trial limitation is likely related to the use of a historical control population as a control group, including patients followed in two different periods (with potential differences in the clinical monitoring). Another limitation is related to the sample size, which impedes the statistical significance reached in a comparative model: however, although the group × time interaction analyzed with a linear mixed factor analysis was not statistically significant, even though a clear trend is evident-for the weight and ALSFRS-R progression in this small pilot study, the findings on weight are extremely encouraging for conceiving a potential benefit for nutritional counseling and redefining a randomized, in-parallel, clinical trial. Lastly, the ability to manage technological devices is one inclusion criteria: this point can generate an inevitable selection bias.

## 5. Conclusions

After the feasibility and usability pilot study has been carried out, we can conclude that the results obtained are auspicious. The Chatbot is beneficial for patients, caregivers, and healthcare providers; it is also easily usable in patients with different functional statuses, and adherence monitoring shows that this telehealth intervention is feasible. Frequent monitoring and advice on nutritional management of ALS patients may be beneficial in reducing weight loss. Furthermore, collecting and storing data through the conversational system can become standardized monitoring for the nutritional aspects of ALS patients’ during their daily routines.

## 6. Take Home Points

-Weight loss and malnutrition are major obstacles in ALS management and course. Regular nutritional assessment and appropriate interventions are crucial within a multidisciplinary management.-Nutritional intervention should include a flexible dietary plan, adjustable in accordance with disease complications (dysphagia) and clinical needs (energy and protein requirements).-A telehealth approach is a feasible and useful tool in ALS population in order to provide effective nutritional monitoring and advice, early detecting swallowing disorders and weight loss and adjusting nutritional strategy.

## Figures and Tables

**Figure 1 jcm-11-04286-f001:**
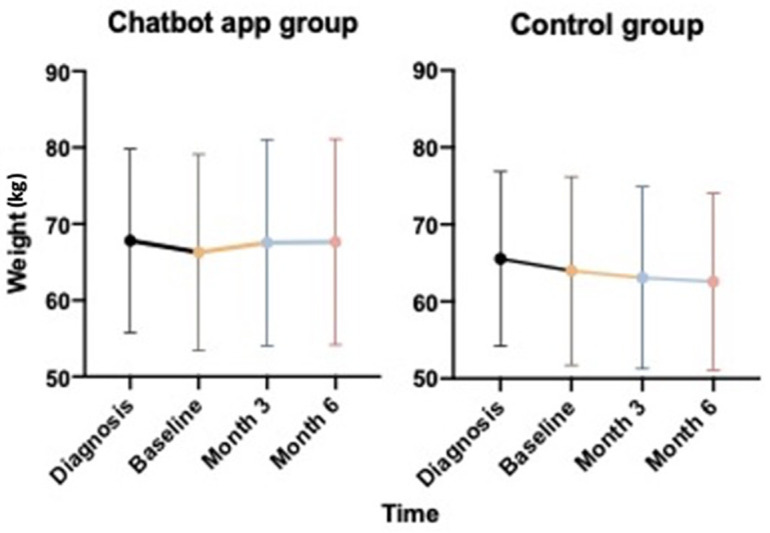
Weight evolution over disease course with (**left** panel) and without (**right** panel) the Chatbot use: intra-rate analysis.

**Figure 2 jcm-11-04286-f002:**
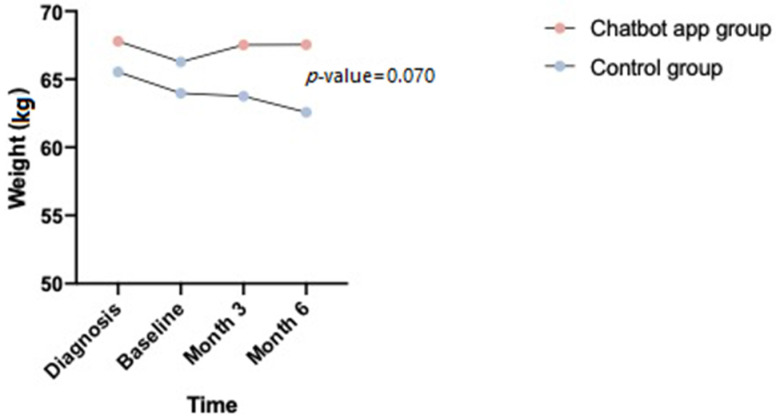
Weight evolution over disease course: inter-rate analysis.

**Figure 3 jcm-11-04286-f003:**
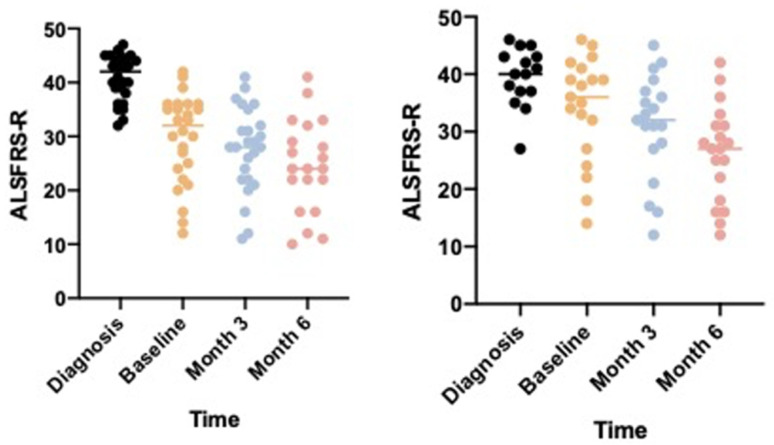
ALSFRS-R evolution over disease course with (panel on the **right**) and without (panel on the **left**) Chatbot use. ALSFRS-R: Amyotrophic Lateral Sclerosis Functional Rating Scale-Revised.

**Figure 4 jcm-11-04286-f004:**
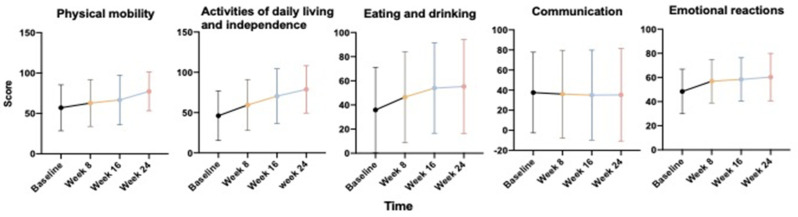
Quality of life during the dietary intervention collected by the ALSAQ-40 scale (five items).

**Table 1 jcm-11-04286-t001:** Demographic and functional features at diagnosis and trial enrollment. No significant differences were observed between the two groups at diagnosis and baseline (*p*-value > 0.05 in all variables). Abbreviations: SD: standard deviation; ALSFRS-R: Amyotrophic Lateral Sclerosis Functional Rating Scale-Revised.

	Chatbot Care	Standard of Care	*p*-Value
Male/female (*n*, %)	17/9 (65%/35%)	10/10 (50%/50%)	0.293
Age at onset (years, SD)	57.2 (13.6)	59.8 (6.50)	0.435
Bulbar/spinal phenotype (*n*, %)	5/21 (19%/81%)	7/13 (35%/65%)	0.227
Educational level (age)	11.5 (3.94)	11.0 (4.00)	0.674
Disease duration (months, SD)	10.5 (7.25)	8.52 (7.80)	0.379
Total ALSFRS-R score at diagnosis (mean, SD); range	40.90 (4.26) 32–47	39.53 (5.05) 27–46	0.233
Total ALSFRS-R score at baseline (mean, SD); range	35.04 (7.74) 13–48	34.05 (9.15) 14–46	0.693
ALSFRS-R score (item 1–3) at baseline (mean, SD)	10.04 (2.07)	8.58 (3.65)	0.093
ALSFRS-R score (item 4–6) at baseline (mean, SD)	7.52 (3.38)	7.21 (3.94)	0.776
ALSFRS-R score (item 7–9) at baseline (mean, SD)	6.80 (3.05)	7.53 (3.69)	0.466
ALSFRS-R score (item 10–12) at baseline (mean, SD)	10.56 (1.88)	10.74 (2.24)	0.768
Weight at diagnosis (mean, SD)	67.80 (12.00)	65.55 (11.30)	0.521
BMI at diagnosis (mean, SD); range	24.55 (3.80) 18.72–35.56	24.86 (4.74) 18.25–38.22	0.806
Weight at baseline (mean, SD)	66.27 (11.97)	63.98 (11.97)	0.523
BMI at baseline (mean, SD); range	23.72 (3.98) 17.82–36.44	24.54 (4.74) 16.75–38.22	0.527

## Data Availability

The corresponding authors will share data upon reasonable request.

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
