# Peer review of "A Telehealth Intervention for Nutritional Counseling in Amyotrophic Lateral Sclerosis Patients"

_jcm, 2022, doi:10.3390/jcm11154286_

Round 1
Reviewer 1 Report
The authors showed that frequent monitoring using the Chatbot application may be more effective than the traditional quarterly ALS center visits for nutritional guidance. This is a very useful study, but the following points could be better organized. Please consider the following points.
1. The description of the paper is somewhat confusing in that the authors used the Chatbot application to monitor and advise ALS patients on their nutritional status, and that the monitoring was more frequent than in the control group.
As the authors state as a limitation, this is a pilot study based on a small number of patients.
The authors suggest that this is a pilot study based on a small number of cases, which may be useful for future research.
Frequent monitoring and advice on nutritional management of ALS patients may be beneficial in reducing weight loss.
The Chatbot is likely to be useful in providing frequent monitoring and advice.
The argument could be better organized to make these two points clearer.
2. There are some spelling errors. Please correct them.
Author Response
We would like to thank the reviewer for careful and thorough reading our paper. Our response follows the reviewer comments, which are in are in italics.
Comment 1 - Reviewer #1: Content:
The description of the paper is somewhat confusing in that the authors used the Chatbot application to monitor and advise ALS patients on their nutritional status, and that the monitoring was more frequent than in the control group.
- As the authors state as a limitation, this is a pilot study based on a small number of patients.
- The authors suggest that this is a pilot study based on a small number of cases, which may be useful for future research.
- Frequent monitoring and advice on nutritional management of ALS patients may be beneficial in reducing weight loss.
- The Chatbot is likely to be useful in providing frequent monitoring and advice.
The argument could be better organized to make these two points clearer.
Reply to reviewer: thanks for the possibility to better integrate these points in the manuscript. We modified the Discussion based on your advice.
Comment 2 - Reviewer #1: Content:
There are some spelling errors. Please correct them.
Reply to reviewer: thanks, we revised the spelling errors.
Reviewer 2 Report
A nice approach to ALS patients nutrition. I have some comments:
1.El Escorial critera - and now for many years Awaji-Shima. They are higher quality and more ALS patients are diagnosed in time
2.Selection patients for electronic communication with doctors/nutrition specialists can be one factor for involuntary selection.
3.ALS-FRS-R - is not detailed enough for communication and changes of emmotions, according to results of Your study. This could be more discussed.
4.6-months followed up period with stabilization in Chatbot group - more facts and ideas could be added.
Author Response
We would like to thank the reviewer for careful and thorough reading our paper. Our response follows the reviewer comments, which are in are in italics.
Comment 1 - Reviewer #2: Content:
El Escorial criteria - and now for many years Awaji-Shima. They are higher quality and more ALS patients are diagnosed in time
Reply to reviewer: we used the El Escorial criteria, and we added the primary reference and the references for the revised criteria.
Comment 2 - Reviewer #2: Content:
Selection patients for electronic communication with doctors/nutrition specialists can be one factor for involuntary selection.
Reply to reviewer: thanks for the possibility to clarify this point. We agree with your observation, and we added this to the Limitation paragraph.
Comment 3 - Reviewer #2: Content:
ALS-FRS-R - is not detailed enough for communication and changes of emmotions, according to results of Your study. This could be more discussed.
Reply to reviewer: yes, in ALS clinical practice, the ALSFRS-R is only a functional scale used to evaluate the cranial, upper limb, lower limb and respiratory district but it is not a scale used for psychological / emotional evaluation. In addition, on regard to the communication, in the ALSFRS-R only one question (1/12) is about the ability to speak, therefore it can be not very specific.
Comment 4 - Reviewer #2: Content:
6-months followed up period with stabilization in Chatbot group - more facts and ideas could be added.
Reply to reviewer: thanks for the possibility to highlight this point. We added a comment in the discussion. Clearly these are only preliminary data on a small cohort of patients, that should be increased.